# RNA Sequencing (RNA-Seq) Analysis Reveals Liver Lipid Metabolism Divergent Adaptive Response to Low- and High-Salinity Stress in Spotted Scat (*Scatophagus argus*)

**DOI:** 10.3390/ani13091503

**Published:** 2023-04-28

**Authors:** Jieqing Chen, Bosheng Cai, Changxu Tian, Dongneng Jiang, Hongjuan Shi, Yang Huang, Chunhua Zhu, Guangli Li, Siping Deng

**Affiliations:** 1Fisheries College, Guangdong Ocean University, Zhanjiang 524088, China; 2Guangdong Research Center on Reproductive Control and Breeding Technology of Indigenous Valuable Fish Species, Zhanjiang 524088, China; 3Guangdong Research Center on Reproductive Control and Breeding Technology of Indigenous Valuable Fish Species, Guangdong Provincial Key Laboratory of Pathogenic Biology and Epidemiology for Aquatic Economic Animals, Fisheries College, Guangdong Ocean University, Zhanjiang 524088, China

**Keywords:** transcriptome, eurihaline fish, osmoregulation, brackish water

## Abstract

**Simple Summary:**

The liver transcriptome was analyzed after 22 d culture with different salinity water using RNA sequencing (RNA-seq) in spotted scat (*Scatophagus argus*). The genes involved in lipid metabolism were differentially down- or up-regulated in low (5 ppt)- and high (35 ppt)-salinity-rated fish in comparison with the control (25 ppt) group. The difference in liver lipid metabolism is important to adapt to low- and high-salinity stress in spotted scat, which has an important value for understanding the molecular basis of salinity adaptation in euryhaline fish.

**Abstract:**

Spotted scat (*Scatophagus argus*) can tolerate a wide range of salinity fluctuations. It is a good model for studying environmental salinity adaptation. Lipid metabolism plays an important role in salinity adaptation in fish. To elucidate the mechanism of lipid metabolism in the osmoregulation, the liver transcriptome was analyzed after 22 d culture with a salinity of 5 ppt (Low-salinity group: LS), 25 ppt (Control group: Ctrl), and 35 ppt (High-salinity group: HS) water by using RNA sequencing (RNA-seq) in spotted scat. RNA-seq analysis showed that 1276 and 2768 differentially expressed genes (DEGs) were identified in the LS vs. Ctrl and HS vs. Ctrl, respectively. Kyoto Encyclopedia of Genes and Genomes (KEGG) analysis showed that the pathways of steroid hormone biosynthesis, steroid biosynthesis, glycerophospholipid metabolism, glycerolipid metabolism, and lipid metabolism were significantly enriched in the LS vs. Ctrl. The genes of steroid biosynthesis (*sqle*, *dhcr7*, and *cyp51a1*), steroid hormone biosynthesis (*ugt2a1*, *ugt2a2*, *ugt2b20*, and *ugt2b31*), and glycerophospholipid metabolism (*cept1*, *pla2g4a*, and *ptdss2*) were significantly down-regulated in the LS vs. Ctrl. The pathways related to lipid metabolisms, such as fatty acid metabolism, fatty acid biosynthesis, peroxisome proliferator-activated receptor (PPAR) signaling pathway, adipocytokine signaling pathway, fatty acid degradation, and unsaturated fatty acid biosynthesis, were significantly enriched in the HS vs. Ctrl. The genes of unsaturated fatty acid biosynthesis (*scd1*, *hacd3*, *fads2*, *pecr*, and *elovl1*) and adipocytokine signaling pathway (*g6pc1*, *socs1*, *socs3*, *adipor2*, *pck1*, and *pparα*) were significantly up-regulated in the HS vs. Ctrl. These results suggest that the difference in liver lipid metabolism is important to adapt to low- and high-salinity stress in spotted scat, which clarifies the molecular regulatory mechanisms of salinity adaptation in euryhaline fish.

## 1. Introduction

Salinity, an important environmental factor, can influence the physiological and metabolic activity of marine fish [1,2]. Euryhaline fish can adapt to a wide range of salinity and have a superior ability for osmoregulation under chronic and rapid osmotic stress [3]. Current reports on salinity adaptation focus mainly on osmoregulation [4,5], and little research has focused on osmoregulation-linked energy metabolism in fish. Studies on energy metabolism in osmotic adjustment were mainly about the physiological response due to osmotic pressure [6,7]. Little is known at the molecular level, such as changes in specific signal pathways and genes.

To maintain the osmotic balance of water and solutes in cells, the cell membrane plays a critical role [8]. It can help to form a semi-permeable barrier to separate the interior of the cell from the inter-cellular environment. As the major lipid components of the cell membrane, phospholipids and cholesterol interact to form a bilayer that affects its permeability and fluidity [9,10]. The genes involved in glycerophospholipid metabolism and steroid biosynthesis are up-regulated in the energy metabolic processes in the liver of coastal fish under low-salinity stress to maintain osmotic homeostasis [11].

As the primary metabolic organ, the liver has been shown to be the primary source of carbohydrate metabolites to osmoregulatory organs [12]. Related research has shown that the liver can promote glycogen decomposition into glucose to maintain normal blood sugar levels and provide energy for osmoregulatory organs under salinity stress in *Sparus aurata* [13,14]. In comparison, osmoregulatory organs such as gills and kidneys have received extensive attention for many years [15,16]. However, the role of the liver in fish osmoregulation is poorly understood. Therefore, research on fish osmoregulation should open a new chapter to study the activities of the liver and how they relate to salinity adaptation.

The spotted scat (*Scatophagus argus*), a euryhaline fish, is an important commercial marine fish in China. Spotted scat is popular among customers because of its good nutrient quality, high protein content, and delicious taste [17]. It can survive in a wide range of salinity, providing an excellent model to study osmoregulation during acclimation to various salinity. Plasma 17*β*-estradiol and 11-ketotestosterone levels increase significantly with increasing salinity acclimation time [18], and better growth and survival in low-salinity (5 ppt) culture [19] have been reported in spotted scat.

Dietary fish oil and starvation re-feeding can influence the lipid metabolism in spotted scat [20]. However, the molecular mechanisms, especially lipid metabolism and the roles of lipid metabolism in salinity adaptation against salinity stress, remain unclear. To better understand the mechanism of salinity adaptation, the relationship between salinity stress and lipid metabolism in livers was investigated using transcriptome sequencing technology. The results help to clarify molecular and biological processes involved in the salinity adaptation in spotted scat.

## 2. Materials and Methods

### 2.1. Animals, Salinity Stress, Fish Sampling, and Ethics Statement

One-year-old spotted scat (body length: 16.9–18.9 cm, body weight: 148.1–211.5 g) were purchased from a farm in Mazhang District, Zhanjiang City, Guangdong Province, China. The acclimation and salinity stress were carried out at Zhanjiang Donghai Island Cultivation Base (Zhanjiang, Guangdong, China).

Before salinity stress, the spotted scat were acclimated for 10 d at 25 ppt salinity, a natural photoperiod, and 28–29 °C temperature. Six cement ponds (5 m × 4 m × 3 m) were used for salinity stress with a salinity of 5 ppt (Low-salinity group: LS), 25 ppt, and 35 ppt (High-salinity group: HS) water, respectively. The salinity of 25 ppt was taken as the control group (Ctrl). There were two repeats for each salinity treatment group. The LS and Ctrl were made with a mixture of saltwater and freshwater, whereas the HS was made with seawater and sea crystal. After acclimation, the spotted scat was transferred to cement ponds with salinity stress for 22 d. Commercial floating meals were provided at 08:00 and 16:00 daily (total feeding rate was roughly 1% of fish body weight) during acclimation and salinity stress. Nine fish were sampled randomly from each group. Three fish were used as a single sample, and three biological replicates were used for sequencing in each group. Liver tissues were frozen in liquid nitrogen before being moved to a −80 °C refrigerator for RNA extraction and reverse transcription. Before sampling, the fish were anesthetized with 300 mg/L tricaine methanesulfonate (MS-222; Sigma-Aldrich, St. Louis, MO, USA). The animal ethics protocol was approved by Guangdong Ocean University’s Animal Research and Ethics Committees (NIH Pub. No. 85-23, amended 1996).

### 2.2. RNA Extraction, Complementary DNA (cDNA) Library Construction, and Sequencing

The total RNAs of the liver were isolated using Trizol reagent (Invitrogen, Waltham, MA, USA) according to the manufacturer’s instructions (Takara, Shiga, Japan). An Agilent 2100 Bioanalyzer (Agilent Technologies, Palo Alto, CA, USA) was used to examine the quality, and RNA integrity was confirmed using agarose gel electrophoresis. Messenger RNA (mRNA) was isolated using magnetic beads. The fragmentation of mRNA was achieved at high temperatures and with the use of metal ions. The cDNA strand was synthesized with random hexamers primers. This process was followed by second-strand cDNA synthesis, end repair, adaptor ligation, fragment sorting, and amplification by PCR. After building the library, the effective concentration (effective library concentration > 2 nM) was determined using reverse transcription quantitative real-time polymerase chain reaction (RT-qPCR). The libraries were sequenced by Biomarker Technologies Ltd. (Beijing, China) using the Illumina HiSeq 2500 platform (Illumina, San Diego, CA, USA). The transcriptome sequencing of LS, HS, and Ctrl was performed in triplicate.

### 2.3. Differential Gene Expression Analysis and Functional Enrichment

Differentially expressed genes (DEGs) were identified using the DESeq2 [21] software package by comparing Ctrl vs. LS and Ctrl vs. HS. The fragments per kilobase of transcript per million mapped reads (FPKM) method was used to estimate the gene expression levels of each transcript. To measure the degree of association between two samples, the sample correlation coefficient was used to quantify the gene FPKM of biological repeat samples. A heat map of FPKM-normalized transcript, which counts for the DEGs in each pairwise comparison, was generated to illustrate the differential expression of genes detected in the liver among different salinities. Genes with |log_2_foldchange| > 0 and *p* < 0.05 were identified as significant DEGs. DEGs were analyzed by the Kyoto Encyclopedia of Genes and Genomes (KEGG) pathway using KOBAS software (http://kobas.cbi.pku.edu.cn/home.do, accessed on 16 March 2021). The selection criterion for enriched KEGG pathways was set at *p* < 0.05.

### 2.4. Data Validation and Expression of Genes Related to Lipid Metabolism

To verify the accuracy of RNA sequencing (RNA-seq) data, 15 DEGs were examined for expression levels by RT-qPCR. The expression levels of eight lipid metabolism-related genes were determined using RT-qPCR and RNA-seq. In the preceding experiments, RNA samples were generated from the LS, Ctrl, and HS (with three replicate samples in each group). Primers were designed based on the assembled transcriptome sequence using the Primer 5.0 software, and these gene primers will be used to detect the expression pattern of DEGs. The primer sequences for RT-qPCR are shown in Table 1.

### 2.5. RT-qPCR and Statistical Analysis

The RT-qPCR reaction system was 20 μL and contained 10 μL PerfectStart^TM^ Green qPCR SuperMix (TransGen Biotech, Beijing, China), 1 μL forward (F) primer, 1 μL reverse (R) primer (10 μmol/L), 7 μL ddH_2_O, and 1 μL cDNA. The reactions were conducted on a Roche Light Cycler 96 (Roche, Switzerland). The reaction was done as follows: 95 °C for 5 min, 95 °C for 30 s, 57 °C for 20 s, and 72 °C for 20 s (fluorescence collection) for 40 cycles. Melting curve analysis was performed to identify the specificity of PCR products. Using *β-actin* as an internal reference gene, the relative expression of each gene in liver tissue was determined by the 2^−ΔΔCt^ calculation technique based on the Ct values obtained by RT-qPCR [22]. Data were expressed as mean ± standard deviation (SD). The Analysis of Variance (ANOVA) test was used to assess the significance of differences between the treatment and control groups by Statistical Product and Service Solutions (SPSS) 19.0 (SPSS, Chicago, IL, USA). Differences were considered significant when *p* < 0.05.

## 3. Results

### 3.1. Illumina Sequencing and Reads Mapping

A total of 19.25 million clean reads were obtained, including 6.43, 6.46, and 6.35 million reads in the LS, HS, and Ctrl, respectively. The correlation coefficients of the sample showed that the replicate samples within the group were highly correlated and stable (Appendix A). The distribution of FPKM showed that log10(FPKM) is mainly distributed from −2 to 2 and presented as a single peak (Appendix A). For each group, 5.98–6.65 GB of clean bases were generated. More than 96.54% and 91.36% of bases had a base accuracy of Q20 and Q30, respectively. A total of 384.03 and 358.86 million clean reads were mapped and uniquely mapped onto the annotated genome of the spotted scat (GenBank accession no.: GCA_020382885.1), respectively (Table 2).

### 3.2. Differential Expression Analysis

A total of 4044 DEGs were identified, with 699 up-regulated and 577 down-regulated genes in the LS vs. Ctrl and 1294 up-regulated and 1474 down-regulated genes in the HS vs. Ctrl (Figure 1A,B). Hierarchical clustering was performed by selecting a joint set of DEGs based on the observed gene expression patterns. Compared with the Ctrl group, there were significant differences in the global gene expression profiles both in the LS group and in the HS group. Furthermore, the gene expression patterns of the three groups of biological replicate individuals could be clustered together separately (Figure 1C,D).

### 3.3. RNA-Seq Validation

The results from RT-qPCR showed that the expression of *igf2* (Insulin-like growth factor II) and pck1 (Phosphoenolpyruvate carboxykinase) were down-regulated. Expression levels of *gstt1a* (Glutathione S-transferase theta-1a), *scd1* (Stearoyl-CoA desaturase 1), *igfbp5* (Insulin-like growth Factor-binding protein 5), *fabp* (Fatty acid-binding protein), *me1* (NADP-dependent malic enzyme), and *lrata* (Lecithin retinol acyltransferase a) were up-regulated compared with the LS vs. Ctrl. The expression levels of *scd1*, *acaca* (Acetyl-CoA carboxylase 1), *acsl4* (Long-chain-fatty-acid--CoA ligase 4), *slc4a4a* (Electrogenic sodium bicarbonate cotransporter 1), *slc25a25a* (Calcium-binding mitochondrial carrier protein SCaMC-2), *arl4aa* (ADP-ribosylation factor-like protein 4A), and *me1* genes were up-regulated in the HS vs. Ctrl. The expression patterns of the 15 DEGs were in agreement with the results of RNA-seq (Figure 2).

### 3.4. KEGG Enrichment Analysis

A total of 2297 DEGs were annotated to the KEGG pathway. According to their functions, the KEGG pathways were classified into material metabolism (Carbohydrate metabolism, Protein metabolism, Amino acid metabolism, Lipid metabolism, and Metabolism of other amino acids), Immune system, Transcription and translation, Energy metabolism, Transport and catabolism, and Signal transduction and Growth (Figure 3, Appendix A).

In the LS vs. Ctrl, a total of 895 DEGs were annotated to 151 pathways, and 27 KEGG pathways were significantly enriched (*p* < 0.05) (Figure 3A). These enriched pathways are related to Material metabolism, Energy metabolism, Immune system, Growth, Transcription and translation, and Signal transduction. Ten enriched pathways involved in the Immune system, Transcription and translation, material metabolism, Energy metabolism, and Growth processes were up-regulated. Seventeen enriched pathways involved in the Immune system, Energy metabolism, Transcription and translation, Growth, and Signal transduction were down-regulated. There was a down-regulation trend of pathways involved in lipid metabolism, including steroid hormone biosynthesis, steroid biosynthesis, glycerophospholipid metabolism, and glycerolipid metabolism.

In the HS vs. Ctrl, 1402 DEGs were annotated to 158 pathways, and 39 pathways were significantly enriched. Twenty-nine pathways were up-regulated, and 10 pathways were down-regulated in the HS vs. Ctrl. A total of 72.41% (21/29) of the up-regulated pathways were associated with material metabolism. Of these, there was an up-regulation trend of pathways associated with lipid metabolism, including fatty acid metabolism, fatty acid biosynthesis, peroxisome proliferator-activated receptor (PPAR) signaling pathway, adipocytokine signaling pathway, fatty acid degradation, biosynthesis of unsaturated fatty acids, and synthesis and degradation of ketone bodies (Figure 3B).

### 3.5. Expression Levels of Lipid Metabolism-Related Genes

The results from RT-qPCR and RNA-seq showed that the expression levels of *fas* (Fatty acid synthase), *hl* (Hepatic lipase), *scd1*, *acc* (Acetyl-CoA carboxylase 1), *pparα* (Peroxisome proliferator-activated receptor α), and *aco* (Acyl-CoA oxidase) were up-regulated, while there was a down-regulation of *cpt1* (Carnitine palmitoyltransferase 1) and no significant difference of *lpl* (Lipoprotein lipase) in the HS vs. Ctrl. The expression levels of *hl*, *acc*, *pparα*, and *aco* were not changed, with *fas* and *scd1* up-regulation and *lpl* and *cpt1* down-regulation in LS vs. Ctrl (Figure 4). These results indicated that the expression patterns of lipid metabolism-related genes detected by RT-qPCR were consistent with the results of RNA-seq, and the results of reproducibility as well. Compared to the control group, low-salinity exposure led to the down-regulation of lipid metabolism-related DEGs, including *sqle*, *dhcr7*, *cyp51a1*, *cept1*, *pla2g4a*, *dgki*, *ptdss2*, *dgkd*, *pck1,* and *ugt* family genes. In contrast, high-salinity exposure up-regulated the expression of *pecr*, *elovl1*, *fads2*, *scd1*, *fas*, *acaca*, *g6pc1*, *socs1*, *socs3*, *adipor2*, *pck1,* and *pparα* genes (Table 2 and Table 3).

## 4. Discussion

Salinity is an important factor that can affect metabolism in fish [23]. Lipids play an important role in fish salinity adaptation by acting as energy donors in energy metabolism and regulators in osmoregulation [24]. The results of FPKM analysis showed that the biological repeat samples within the group were highly correlated and stable. The gene expression results detected by RT-qPCR showed that the RNA-seq results were highly reproducible and the RNA-seq sequencing data were accurate. A total of 1276 DEGs (699 up-regulated and 577 down-regulated) in the LS vs. Ctrl and 2768 DEGs (1294 up-regulated and 1474 down-regulated) in the HS vs. Ctrl were identified. KEGG analysis showed that the pathways involved in lipid metabolism, including steroid hormone biosynthesis, steroid biosynthesis, glycerophospholipid metabolism, and glycerolipid metabolism, were significantly enriched in the LS vs. Ctrl. The pathways associated with lipid metabolism, including fatty acid metabolism, fatty acid biosynthesis, PPAR signaling pathway, adipocytokine signaling pathway, fatty acid degradation, biosynthesis of unsaturated fatty acids, and synthesis and degradation of ketone bodies, were significantly enriched in the HS vs. Ctrl, which suggests that lipid metabolism plays an important regulatory role in salinity adaptation in spotted scat.

### 4.1. Steroid Biosynthesis

Steroids play a key role in response to salinity changes in aquatic organisms [25]. In the present study, KEGG analysis showed that the steroid biosynthesis pathway was significantly enriched. RNA-seq showed that the genes associated with this pathway, such as *sqle* (Squalene epoxidase), *dhcr7* (7-dehydrocholesterol reductase), and *cyp51a1* (Lanosterol 14 alpha-demethylase), were significantly down-regulated in the LS vs. Ctrl. However, the expression levels of these genes were not changed in the HS vs. Ctrl. Sterols are important compounds of biological membranes; they are components of cell membranes and have important roles in transport capability [11]. Sqle catalyzes the first oxygenation step in sterol biosynthesis and is the rate-limiting enzyme for cholesterol biosynthesis in this pathway [26]. Lanosterol is an upstream precursor for the biosynthesis of animal and fungal steroid sterols, especially cholesterol [27]. Cyp51a1 catalyzes the 14-alpha demethylation of lanosterol and the eventual formation of cholesterol through a complex reaction [28]. Dhcr7, a membrane-bound enzyme, catalyzes the final step of cholesterol biosynthesis using nicotinamide adenine dinucleotide phosphate (NADPH) [29]. In this study, *sqle*, *dhcr7*, and *cyp51a1* were significantly down-regulated in the LS vs. Ctrl, which suggests that cholesterol levels are reduced in the liver under low-salinity conditions in spotted scat. Cholesterol is an important lipid component. It is also an essential part of cell membranes [30]. Cholesterol biosynthesis plays a crucial role in the relationship between osmoregulation and steroids, and the cholesterol induced by salinity fluctuations is related to the physical properties of cell membranes [31]. For instance, increasing cholesterol intake can improve the osmotic pressure regulation and environmental adaptability of white shrimp [32]. In summary, the results indicate that steroid biosynthesis is involved in osmoregulation. Reducing cholesterol biosynthesis is important in maintaining osmoregulation homeostasis under low-salinity conditions in spotted scat.

### 4.2. Steroid Hormone Biosynthesis

UDP-glucuronosyltransferase can convert small lipophilic molecules, such as steroid hormones, into water-soluble excretable metabolites [33,34]. In this study, the pathway of the steroid hormone biosynthesis was significantly enriched, and several UDP-glucuronosyltransferase (UGT) family genes (*ugt2a1*, *ugt2a2*, *ugt2b20*, and *ugt2b31*) associated with this pathway were significantly down-regulated in the LS vs. Ctrl. However, the expression levels of these genes were not changed significantly in HS vs. Ctrl. Glucuronidation, catalyzed by UGTs, is a key pathway in the metabolism and homeostasis of endogenous molecules and plays a key role in the detoxification and excretion of contaminants in fish [35]. The down-regulation of UGTs genes affects organismal homeostasis under hypoxic stress [36], and stilbene glucoside inhibits the metabolism of emodin in rats by down-regulating the expression of the UTG gene [37]. In addition, starvation induces the up-regulation of UGTs gene expression in mouse liver, which plays an important role in regulating hormone levels and eliminating harmful substances [38]. The down-regulation of these *ugts* genes indicates that it was important to adaptation to low-salinity by reducing the metabolism of small lipophilic molecules, such as steroid hormones in LS vs. Ctrl of spotted scat.

### 4.3. Glycerophospholipid Metabolism

Glycerophospholipids are abundant cellular lipids with important physiological roles in membrane structure (phospholipids and other lipids, depending on the organelle type) and energy storage (mainly in the form of triacylglycerols) [39]. In this study, the glycerophospholipid metabolic pathway was significantly enriched in the LS vs. Ctrl. Cept1 (Choline/ethanolamine phosphotransferase 1), Pla2g4a (Cytosolic phospholipase A2), and Ptdss2 (Phosphatidylserine synthase 2) were involved in this pathway. They were changed under salinity stress in spotted scat. Pla2g4a, a member of the phospholipase A2 family with calcium-dependent phospholipase and lysophospholipase activities, plays a major role in the biosynthesis of lipid mediators for membrane phospholipid remodeling and inflammatory responses [40]. Impaired phospholipid remodeling can affect the structure, function, and membrane protein activity of cell membranes, which can cause disruptions in signaling cascade responses and may have adverse effects on normal metabolism [41]. Based on RNA-Seq, the expression of the *pla2g4a* gene was down-regulated under high- and low-salinity stress in spotted scat, which suggested that phospholipid remodeling was decreased, and the normal metabolic activities of the cells might be affected under salinity stress. Cept1 is the key enzyme that catalyzes phosphatidylethanolamine biosynthesis from CDP-ethanolamine [42]. After phosphatidylethanolamine synthesis, the Ptdss2 enzyme catalyzes the conversion of phosphatidylethanolamine to phosphatidylserine [43]. Phosphatidylserine is a structural membrane phospholipid that plays a role in cell signaling, blood coagulation, and apoptosis [44]. In this study, high- and low-salinity stresses down-regulated the expression levels of *cept1 and pla2g4a*, and low-salinity stress down-regulated *ptdss2* mRNA, which indicated that phosphatidylserine conversion efficiency was reduced during salinity stress. The results of glycerophospholipid metabolism analysis indicated that cell membrane structure and energy storage might be changed to adapt to different salinity conditions in spotted scat.

### 4.4. Biosynthesis of Unsaturated Fatty Acids

In the present study, the pathway of biosynthesis of unsaturated fatty acids was significantly enriched in the HS vs. Ctrl. DEGs of *scd1* (Stearoyl-CoA desaturase), *fads2* (Fatty Acid Desaturase 2), *pecr* (Peroxisomal trans-2-enoyl-CoA reductase), and *elovl1* (Polyunsaturated fatty acid elongase) were found under salinity stress in this pathway. Scd1 catalyzes the insertion of a cis double bond into the δ-9 position of fatty acyl-CoA substrates (Palmitoyl-CoA and Stearoyl-CoA). It is the rate-limiting enzyme for the synthesis of palmitoleic (C 16:1) and oleic (C 18:1) unsaturated fatty acids [45]. Palmitoleic acid can reduce the effects of insulin resistance [40]. Insulin resistance decreases the efficiency of glucose uptake and utilization and affects the metabolism of the body [46]. The levels of *scd1* were up-regulated in the LS vs. Ctrl and in the HS vs. Ctrl, which suggested that palmitoleic acid synthesis and metabolism of steatosis and energy were increased under salinity stress in spotted scat. The intracellular triglyceride content of liver cells increased with increasing oleic acid concentration, which suggests that oleic acid enhances hepatocyte triglyceride synthesis [47]. Triglyceride is an important energy source for fish under environmental stress [48,49]. The triglyceride content of fish significantly decreased with increasing starvation stress time [50]. Delta (6) fatty acid desaturase, encoded by the *fads2* gene, is the rate-limiting enzyme that catalyzes the conversion of linoleic acid to arachidonic acid. In animals, arachidonic acid controls cell membrane fluidity and can indirectly act in cellular signaling processes, such as inflammatory response mediation and synaptic transmission [51]. In the present study, *fads2* expression was up-regulated in the HS vs. Ctrl, indicating that a high-salinity environment contributes to spotted scat’s anabolism of arachidonic acid. Overall, the DEGs involved in the biosynthesis of unsaturated fatty acids pathways suggested that they can alter their membrane structure and energy storage to adapt to the salinity stress in spotted scat. The increase in unsaturated fatty acids may provide an energy source, modulation cell membrane fluidity, and signaling processes. The decrease in phospholipid remodeling may affect the normal metabolic activities of the cells.

### 4.5. Adipocytokine Signaling Pathway

Adipocytokines play important roles in energy homeostasis, immune regulation, and lipid metabolism by acting on target organs through autocrine or paracrine effects and blood circulation [52]. In this study, the adipocytokine signaling pathway was significantly enriched. Genes of *g6pc1* (Glucose-6-phosphatase 1), *socs1* (Suppressor of cytokine signaling 1), *socs3* (Suppressor of cytokine signaling 3), *adipor2* (Adiponectin receptor protein 2), *pck1* (Phosphoenolpyruvate carboxykinase 1), and *pparα* (Peroxisome proliferator-activated receptor alpha) in this pathway were significantly changed in the HS vs. Ctrl. Of these genes, the expression level of the *pck1* gene was down-regulated in the LS vs. Ctrl. Adipor2 is an important hormone secreted by adipocytes to regulate glucose and lipid metabolism [53]. In combination with adiponectin, it activates a signaling cascade that increases PPARα activity and accelerates fatty acid oxidation and glucose uptake [54,55]. PPARα is a key regulator of lipid metabolism, regulating the *β*-oxidation pathway of fatty acids, and fatty acid *β*-oxidation can supply a large amount of the energy that the organism requires [56]. After high-salinity stress, the expression levels of *adipor2* and *pparα* were up-regulated, indicating an increase in energy consumption to adapt to the hyperosmotic environment in spotted scat. High-salinity stress up-regulated the expression levels of glucose synthesis-related genes, including *g6pc1* and *pck1*. G6pc1 is a key enzyme in the homeostatic regulation of blood glucose levels. It forms a complex with the glucose-6-phosphate transporter responsible for glucose production in the final step of glycogenolysis and gluconeogenesis [57]. Pck1 is the rate-limiting enzyme for gluconeogenesis, catalyzing the conversion of oxaloacetate to phosphoenolpyruvate to produce glucose from lactate and other precursors from the citric acid cycle [58,59]. *G6pc1* and *pck1* expression were up-regulated under high-salinity stress. However, the *pck1* expression was down-regulated in the LS vs. Ctrl. To keep energy homeostasis, an increase and decrease in glucose synthesis may be one of the strategies to adapt hyperosmotic and hypotonicity environments in spotted scat.

The immune response is mainly controlled by specific cytokines, such as interleukin 17 (IL-17) and interleukin 22 (IL-22) [60]. The suppressor of the cytokine signaling (SOCS) family plays a negative regulatory role in regulating cytokine signaling pathways related to immunity, growth, and development [61]. In the present study, high-salinity stress significantly up-regulated the expression levels of *socs1* and *socs3* genes. Socs1 and Socs3 are mainly involved in regulating the toll-like receptor signaling pathway [62]. Studies have reported that fish Socs1 and Socs3 can be up-regulated by lipopolysaccharide stimulation; therefore, Socs1 and Socs3 play an important role in the fish’s innate immune response [63,64]. Moreover, Socs1 can negatively regulate the clearance ability of mouse T cells against mycobacteria [65]. In addition, Socs3 can enhance the defense of cows *Mycobacterium avium* subspecies *paratuberculosis* virus infection [66]. RNA-seq showed that *socs1* and *socs3* were up-regulated in the HS vs. Ctrl, which indicates that liver adipocytokines are involved in immune regulation, especially in inflammatory regulation, during high-salinity stress in the spotted scat.

## 5. Conclusions

The relationship between salinity stress and lipid metabolism in liver was investigated by RNA-seq in spotted scat. The results indicate that low-salinity stress can inhibit steroid hormone metabolism by reducing several *ugts* genes expression, and the reduction of cholesterol biosynthesis plays an important role in maintaining osmoregulation homeostasis under low-salinity conditions. Glycerophospholipid metabolism was involved in salinity adaptation during high- and low-salinity stress. The increase in palmitoleic and oleic acid levels is involved in osmoregulation under low- and high-salinity stress. An increase in arachidonic acid is involved in adapting to high-salinity stress. To keep energy homeostasis, the rate of glucose synthesis was accelerated in the hyperosmotic but reduced in the hypotonicity environment. The liver adipocytokines are involved in immune regulation during high-salinity stress. These results suggest that the difference in liver lipid metabolism is important to adapt to low- and high-salinity stress in spotted scat. It will provide insight to clarify the molecular regulatory mechanisms of salinity adaptation in euryhaline fish.

## Figures and Tables

**Figure 1 animals-13-01503-f001:**
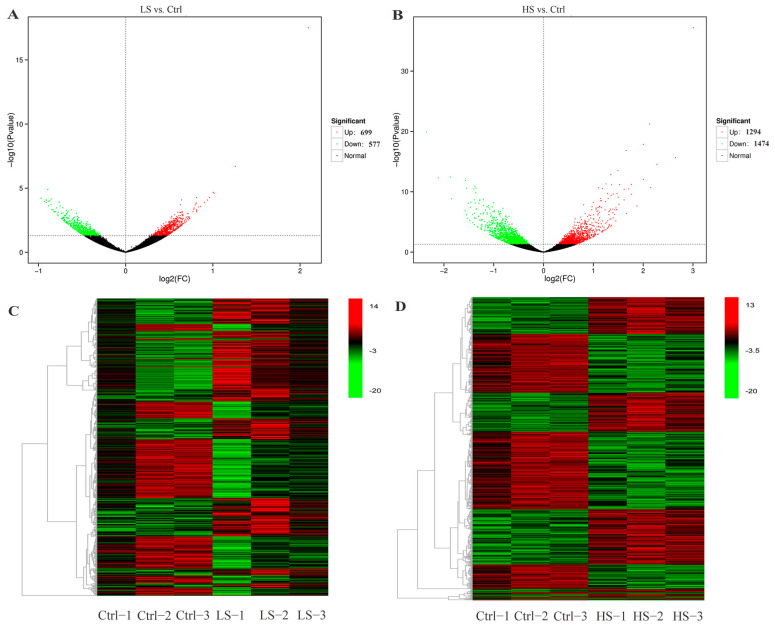
Differential gene expression in the liver challenge with low- and high-salinity in spotted scat. Volcano plot showing the distribution of DEGs salinity challenge in LS vs. Ctrl (**A**) and HS vs. Ctrl (**B**). The heat map summarizes the expression clustering of DEGs with LS vs. Ctrl group (**C**) and HS vs. Ctrl group (**D**). Green and red dots/squares indicate down- and up-regulation, respectively.

**Figure 2 animals-13-01503-f002:**
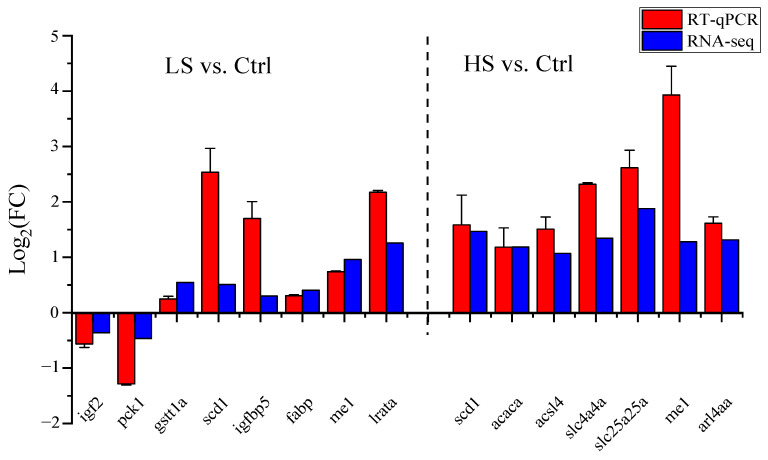
Validation of data accuracy of liver transcriptome challenge with low- and high-salinity by RT-qPCR method in spotted scat. Fold Change (FC): indicates the fold difference in gene expression levels between experimental and control groups.

**Figure 3 animals-13-01503-f003:**
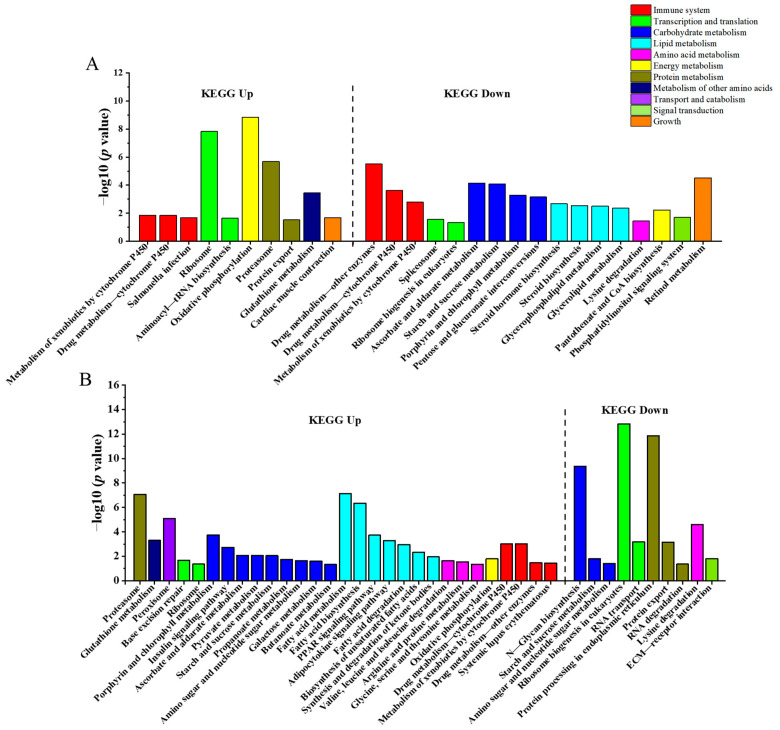
KEGG pathways were identified in the liver challenge with LS vs. Ctrl (**A**) and HS vs. Ctrl (**B**) in spotted scat. The left and right of the black dashed line are the up- and down-regulated KEGG pathways. The vertical axis indicates the −log10 (*p* value) for significant pathways and the horizontal axis indicates the pathway categories.

**Figure 4 animals-13-01503-f004:**
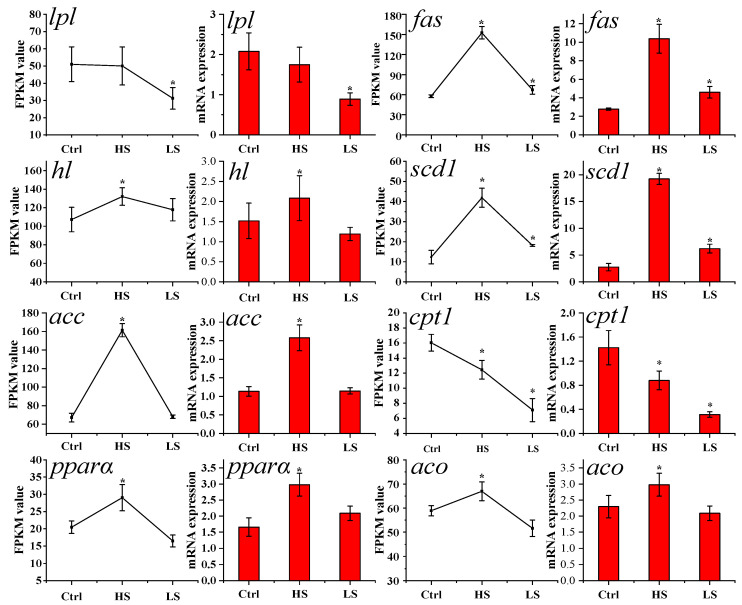
Expression levels of genes related to lipid metabolism in the liver challenge with low- and high-salinity by RT-qPCR and FPKM in spotted scat. FPKM value indicates the gene expression level obtained by RNA-Seq. Data were expressed as mean ± standard deviation. The significance of differences between the treatment and control groups was analyzed by ANOVA (*, *p* < 0.05). Differences between the treatment and control groups were considered significant when *p <* 0.05 and were indicated by an asterisk.

**Table 1 animals-13-01503-t001:** Primer sequences used for RT-qPCR.

Genes Name	Forward Primer (5′-3′)	Reverse Primer (5′-3′)	Application
*β-actin*	GAGAGGTTCCGTTGCCCAGAG	CAGACAGCACAGTGTTGGCGT	validation, lipid metabolism
*lpl*	GCCAATCAAACTGTTGCCAAAT	ACATACCACTCAGCGTCCATCC	lipid metabolism
*fas*	ACTCCTTGATGGGCGTTGA	TGGCTCTGTGCCGTGTTG	lipid metabolism
*hl*	GGAGGAGACTTCCAACCAGG	AGAGTTTGTAAGGCATCCGAGA	lipid metabolism
*scd1*	GTGGCTGCTGGTGCGTAAA	ACCGTCTCGTGGGCAACTC	validation, lipid metabolism
*acc*	GTTGAAGTTCGGGGCCTAC	TGACGAGGGTTGATGGTTGG	validation, lipid metabolism
*cpt1*	GCCGTGGTAAGAACAAGCAAT	CAGTGACCCTCCTCAGTATAGCC	lipid metabolism
*pparα*	TGCCAATACTGCCGCTTCC	GTGCTGGTCTTGCCCGTGA	lipid metabolism
*aco*	CATTGCGGACCCAGAAG	CGGTGGGACTGTTCAAGAC	lipid metabolism
*igf2*	GAGCAGCAGAATGAAGGTC	TCTGCCGCACCTCGTATT	validation
*igfbp5*	TACAAACCTGCCCACCCG	CGCTGCTTGCCTTGCTTC	validation
*gstt1a*	CTCTACAGGGCAGCATTACA	CTCTAGCAGGTTGAGCGACT	validation
*fabp*	AGACCACGCCTGATGACC	TGCCTGGACTCCCTCAAA	validation
*me1*	GCCTCCTTGTTTCATCAGTC	AAAGTTTCTCGTTGTGGTCC	validation
*lrata*	GCCTGCTGTGGAATAACTGTGAA	AAGGGAATAAGGATTGTGGGTAA	validation
*acsl4*	CTATCTGCCTCTGGCTCACG	GTCCGCAACCTCTGTAATGG	validation
*slc4a4a*	TGAGCGAGAACACTTCGGAC	TCGAACTTGATCCACCTGGC	validation
*slc25a25a*	CTGTGGGTTCGTGGAGTC	CTGCCAAGTATTCTTTAGGG	validation
*arl4a*	TTCACATCGCCATCCTCG	CTCCCGCAGTCCATCTCC	validation

**Table 2 animals-13-01503-t002:** Summary of reads of liver transcriptome sequencing challenge with low- and high-salinity in spotted scat.

Group	Ctrl			LS			HS		
Sample name ^a^	25ppt-1	25ppt-2	25ppt-3	5ppt-1	5ppt-2	5ppt-3	35ppt-1	35ppt-2	35ppt-3
Clean reads (×10^6^)	22.00	20.07	21.46	21.31	21.57	21.46	20.90	21.42	22.31
Clean bases (G)	6.56	5.98	6.39	6.36	6.43	6.41	6.21	6.40	6.65
Q20 (%) ^b^	96.8	96.78	96.83	96.79	96.68	96.75	96.58	96.54	96.63
Q30 (%) ^c^	91.92	91.78	91.86	91.90	91.65	91.79	91.41	91.36	91.50
Total mapped (×10^6^) ^d^	43.95	40.17	42.92	42.63	42.13	42.91	41.79	42.85	44.63
Mapping rate (%)	87.68%	88.46%	88.18%	88.39%	87.14%	87.92%	87.92%	87.15%	87.82%
Uniquely mapped (×10^6^)	36.83	34.04	36.32	35.89	36.10	36.08	35.19	35.74	37.48
Uniquely mapped rate (%)	83.82%	84.82%	84.63%	84.19%	83.68%	84.09%	84.21%	83.42%	83.97%

^a^ 1, 2, and 3: three parallel samples; ^b^ Q20: The percentage of bases with a Clean Data quality value greater than 20; ^c^ Q30: The percentage of bases with a Clean Data quality value greater than 30; ^d^ The number of clean reads that mapped onto the reference transcriptome.

**Table 3 animals-13-01503-t003:** DEGs related to liver lipid metabolism challenge with low- and high-salinity stress in spotted scat.

Gene ID ^a^	Gene Name	Log_2_ FC	*p* Value	Gene Function
LS vs. Ctrl				
EVM0003401	*ugt2a1*, UDP-glucuronosyltransferase 2A1	−0.552813189	4.90 × 10^−3^	Steroid hormone biosynthesis
EVM0003565	*ugt2a2*, UDP-glucuronosyltransferase 2A2-like	−0.466258335	3.22 × 10^−2^	
EVM0005100	*ugt2b31*, UDP-glucuronosyltransferase 2B31-like	−0.871147633	9.77 × 10^−5^	
EVM0019647	*ugt2b20*, UDP-glucuronosyltransferase 2B20-like	−0.457260075	2.07 × 10^−2^	
EVM0006384	*sqle*; squalene epoxidase	−0.622426552	7.39 × 10^−3^	Steroid biosynthesis
EVM0010040	*dhcr7*; 7-dehydrocholesterol reductase	−0.507776896	2.00 × 10^−2^	
EVM0023884	*cyp51a1*, lanosterol 14-alpha demethylase	−0.625469837	9.21 × 10^−3^	
EVM0000521	*cept1*; choline/ethanolamine phosphotransferase 1	−0.53137678	1.67 × 10^−2^	Glycerophospholipid metabolism
EVM0012014	*pla2g4a*, phospholipase A2-like	−0.802009596	6.68 × 10^−4^	
EVM0018902	*dgki*; diacylglycerol kinase iota	−0.632031625	8.94 × 10^−3^	
EVM0022818	*ptdss2*; phosphatidylserine synthase 2	−0.379058107	3.49 × 10^−2^	
EVM0006432	*dgkd*; diacylglycerol kinase delta-like	−0.990811261	2.75 × 10^−4^	Glycerolipid metabolism
EVM0000633	*scd1*, delta-9-desaturase 1	0.558057248	6.52 × 10^−3^	Biosynthesis of unsaturated fatty acids
EVM0009831	*pck1*; phosphoenolpyruvate carboxykinase 1	−0.466918319	3.95 × 10^−2^	Adipocytokine signaling pathway
HS vs. Ctrl				
EVM0006069	*pecr*; peroxisomal trans-2-enoyl-CoA reductase	0.381801622	4.13 × 10^−8^	Biosynthesis of unsaturated fatty acids
EVM0002080	*e*lovl1; ELOVL fatty acid elongase 1	0.451575173	6.64 × 10^−3^	
EVM0020874	*fads2*, fatty acid desaturase 2-like	0.938107743	1.88 × 10^−4^	
EVM0000633	*scd1*, delta-9-desaturase 1	1.468072624	9.61 × 10^−8^	
EVM0009139	*fas*, fatty acid synthase	1.243001461	4.06 × 10^−9^	Fatty acid biosynthesis
EVM0023488	*acaca*; acetyl-CoA carboxylase alpha	1.185429522	2.97 × 10^−11^	
EVM0001796	*g6pc1*, glucose-6-phosphatase	1.173745732	4.81 × 10^−4^	Adipocytokine signaling pathway
EVM0001859	*socs1*, suppressor of cytokine signaling	1.179125942	4.51 × 10^−4^	
EVM0003316	*socs3*, suppressor of cytokine signaling 3	1.313341782	1.90 × 10^−7^	
EVM0006476	*adipor2*, adiponectin receptor protein 2	0.430474726	8.40 × 10^−3^	
EVM0009831	*pck1*; phosphoenolpyruvate carboxykinase 1	0.518387425	4.80 × 10^−2^	
EVM0021145	*ppara*; peroxisome proliferator activated receptor alpha	0.481586273	4.83 × 10^−2^	
EVM0000521	*cept1*; choline/ethanolamine phosphotransferase 1	−0.709767922	6.67 × 10^−3^	Glycerophospholipid metabolism (ko00564)
EVM0012014	*pla2g4a*; phospholipase A2-like	−0.548207575	4.77 × 10^−2^	

^a^ Gene ID: was assigned with the database GCA_020382885.1 of spotted scat.

## Data Availability

The raw data of Illumina transcriptome have been submitted in the SRA under accession number PRJNA923112.

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
