# Peer review of "RNA Sequencing (RNA-Seq) Analysis Reveals Liver Lipid Metabolism Divergent Adaptive Response to Low- and High-Salinity Stress in Spotted Scat (Scatophagus argus)"

_animals, 2023, doi:10.3390/ani13091503_

Round 1
Author Response
Response to Reviewer 1 Comments
Point 1:
(1.Introduction)
The reason why the authors focused on lipid metabolism seems to be abrupt.The authors were interested in lipid metabolism since it was major energy source for salinity adaptation (L53-58). On the other hand, in 4. Discussion, compared with energy metabolism, the genes for cellular membrane structure and function were more extensively discussed. It will be reasonable that salinity affect the cellular membrane structure, function, damage and recovery which will be accompanied with metabolic changes in lipids.
So, we propose the authors to add some information and logics on the relationships between cellular membrane and lipid metabolism in osmoregulation in 1.Introduction.
Response: Thank you very much for your suggestion. The roles of cell membrane and lipid metabolism in osmoregulation were added in the Introduction.
L54-L66: Cell membranes play a critical role in maintaining the osmotic balance of water and solutes in cell, with one of important aspect in regulation of lipid metabolism [8]. Lipids is an important components of cell membranes and it can help to form a semi-permeable barrier to separate the interior of the cell from the internal environment. Phospholipids is the major lipid component in the cell membrane. It form a bilayer that affects the permeability of the cell membrane. The presence of sphingolipids and cholestero in cell membranen can affect the fluidity of the membrane as well as its permeability to water [9, 10]. In addition to the role in the formation of cell membranes, lipids also plays an important role in osmoregulation as a source of energy for the cell. Lipid metabolism is involved in the breakdown of fats and other lipids into small molecules to produce energy which can be used by cells. To maintain osmotic pressure homeostasis, steroid biosynthesis and glycerolipid metabolism are involved in the energy metabolic processes under low salinity stress in liver in tongue Sole (Cynoglossus semilaevis) [11].
Point 2:
(Figure organization)
The Fig.3 is just a validation (?). if so, it is a part of the explanation of the experimental process like Table2 and Fig.1. Instead, the Table 3 contains more important information which close to the Fig.2 and 4. Discussion.So we propose the authors to replace the position of Fig.3 and Table 3 in the text and story.
Response: Thank you very much for your suggestion. The position of Figure 3 and Table 3 were replaced in the revised manuscript.
Point 3:
(Figure4)
Some genes used in this experiment were not appeared in Table3. We guess the author spicked up these genes since they were major members for lipid metabolism, without regard to the results of RNA-seq. lf so, this experiment provided, not direct data of RNAseq but “supporting data” for the metabolic response. The objective of this experiment should be explained.
Response: Thank you very much for your comment. Pecr, cept1 and pla2g4a genes did not listed in Table 3, it was a negligence in our previous work. They were added in Table 3.
Point 4:
(4.Discussion)
Not only the response of each gene, overall physiological response should be discussed.Please try to discuss the physiological interpretation of metabolic response by gene expression changes and differences between in LS and HS
Response: Thank you very much for your suggestion. We discussed some of them in the revised manuscript.
L325-L327: The down-regulation of these ugts genes indicated it was important to adaption of low salinity by reducing the metabolism of small lipophilic molecules such as steroid hormones in spotted scat in LS vs. Ctrl.
L352-L354: The results of glycerophospholipid metabolism analysis indicated that cell membrane structure and energy storage might be changed to adapt different salinity conditions in spotted scat.
L378-L382: Overall, the DEGs involved in biosynthesis of unsaturated fatty acids pathways suggested that it can alter its membrane structure and energy storage to adapt the salinity stress in spotted scat. The increase of unsaturated fatty acids may provide energy source, modulation cell membrane fluidity and signaling processes. while the decrease of phospholipid remodeling may affect normal metabolic activities of the cells.
Point 5:
(Table1)
Indicate the experiment or figure number in which each primer used.
Response: Thank you for your suggestion. We listed the information of primers in table 1.
Point 6:
(Fig.3 and Fig.4)
Explain the reason why the authors chose there genes for these analyses.
Response: In Fig.3, it suggested that lipid metabolism involved in salinity adaptation in spotted scat. And other papers also indicated that lipid metabolism involved in salinity adaptation in teleost. In Fig.4, these genes we chosen were all related to lipid metabolism. Therefore, the study of these genes can provide the understanding of the physiological adaptation mechanisms in spotted scat.
Point 7:
(L354) The title should be corrected.
Line 390: “4.5 Biosynthesis of unsaturated fatty acids” was changed to “4.5Adipocytokine signaling pathway”.
Point 8:
(L322-323, L331) Confirm the results. The HS responses of cept1, pla2g4a were not shown in Table3.The gene pecr was not shown in Table3.
Response: Thank you for your comments. Genes of cept1, pla2g4a and pecr were listed in Table 3.
Point 9:
(Results or Discussion) The gene function “Glycerolipid metabolism (LS)and “Fattyacid biosynthesis(HS) were appeared in Table3 but not referred in the text. Why were they neglected?
Response: Thank you for your comments. We were interested in key genes involved in osmoregulation and energy metabolism. And the roles of other genes needed further studies.

Reviewer 2 Report
The Authors adressed all the question rised by the reviewer. the manuscript quality was improved
Author Response
Thank you very much for your review comments!
Reviewer 3 Report
#Major comment
The authors should check the reproducibility of RNA-seq data. For each of the samples (Ctrl, LS and HS, 9 samples in total), the authors should show the distribution of FPKM, and discuss the correlation coefficient as in Figure S1 in the revised manuscript.
#Minor comments
line 119: Cite the paper of DESeq2.
lines 124 and 128: The authors should mention what kind of statistical tests were used for P-value calculation.
line 153: The authors should describe the correlation coefficient in detail. I do not understand what “more than 0.884” means.
lines 169-172: I do not understand what the authors are trying to argue about Figures 1C and 1D. It is only natural that the gene expression patterns were significantly different between three groups, because significant DEGs were used for the clustering. It seems to me that the authors only confirmed that DESeq2 worked well (i.e. picked out appropriate genes). This part should be totally rephrased or deleted.
line 246: was analysis -> was analyzed
Figure S1: Three groups (Ctrl, LS and HS) are not separated to each other. The authors should discuss this point in the main text, which may be involved in the reproducibility of RNA-seq. If the reproducibility is not good, the authors may argue the importance of RT-qPCR as the further validation, after RNA-seq as the initial rough screening.
Table 3: The authors should explicitly mention the source of gene IDs such as EVMxxxxxxx. Were these predicted form the genome sequence in this study or obtained from any previous study?
Author Response
Response to Reviewer 3 Comments
Point 1:
The authors should check the reproducibility of RNA-seq data. For each of the samples (Ctrl, LS and HS, 9 samples in total), the authors should show the distribution of FPKM, and discuss the correlation coefficient as in Figure S1 in the revised manuscript.
Response: Thank you so much for your suggestion.
L171-L174: We have supplemented these as your request. Corresponding content has been explained and revised in the revised manuscript. The FPKM distribution plot for each sample is shown below.
Point 2:
line 119: Cite the paper of DESeq2.
Response: Thank you for your comment.
Line 129: Papers of DESeq2 have been added to the reference.
Point 3:
lines 124 and 128: The authors should mention what kind of statistical tests were used for P-value calculation.
Response: Thank you for your comment.
Line 162-Line 164: The P-value test method has been supplemented.
Point 4:
line 153: The authors should describe the correlation coefficient in detail. I do not understand what “more than 0.884” means.
Response: Thank you so much for your comment.
Line 169-Line 172: We have discussed the results of the correlation coefficient in detail. “more than 0.884”was deleted. we have made corresponding correction in the text.
The correlation coefficients of sample showed the biological repeat samples within the group were highly correlated and stable (Figure S1). The distribution of FPKM shows that log10(FPKM) is mainly distributed from -2 to 2 and presented as a single peak (Figure S2).
Point 5:
lines 169-172: I do not understand what the authors are trying to argue about Figures 1C and 1D. It is only natural that the gene expression patterns were significantly different between three groups, because significant DEGs were used for the clustering. It seems to me that the authors only confirmed that DESeq2 worked well (i.e. picked out appropriate genes). This part should be totally rephrased or deleted.
Response: Thank you so much for your comment.
Two information were presented in Figure 1C and 1D in this article. Firstly, the levels of differential gene expression can be reflected in the heatmap between the control group and the experimental group. Secondly, the gene expression profiles were also reflected among biological replicate samples. Both reliability and stability of gene expression data can be assessed by comparing the expression variations among biological replicates. A new descriptions of Figure 1C and 1D was added in the revised manuscript.
Line 186-Line 191: Hierarchical clustering is performed by selecting a joint set of DEGs based on the observed gene expression patterns. Compared with the Ctrl group, there is a significant differences of the global gene expression profiles both in the LS group and in the HS group. Furthermore, the gene expression patterns of the three groups of biological replicate individuals could be clustered together separately (Figure 1C and 1D).
Point 6:
line 246: was analysis -> was analyzed
Response: Thank you so much for your comment. We changed “analysis” to “analyzed”.
Point 7:
Figure S1: Three groups (Ctrl, LS and HS) are not separated to each other. The authors should discuss this point in the main text, which may be involved in the reproducibility of RNA-seq. If the reproducibility is not good, the authors may argue the importance of RT-qPCR as the further validation, after RNA-seq as the initial rough screening.
Response: Thank you so much for your comment.
The gene expression profiles of all three biological replicates within the group showed it could be clustered together in Figure 1C and 1D which indicated the good concordance of biological replicates. In addition, log2 (FC) of differentially expressed genes and FPKM of lipid metabolism-related genes were also detected using RT-qPCR. The results showed that the expression patterns of 15 differentially expressed genes and 8 lipid metabolism-related genes detected by RT-qPCR were consistent with the RNA-seq sequencing results. These results suggested the high reproducibility and stability of RNA-seq.
Point 8:
Table 3: The authors should explicitly mention the source of gene IDs such as EVMxxxxxxx. Were these predicted form the genome sequence in this study or obtained from any previous study?
Response: Thank you so much for your comment. Gene IDs were assigned with the genome database GCA_020382885.1 of spotted scat. It was noted in Table 3.
L269: a Gene ID: were assigned with the database GCA_020382885.1 of spotted scat.

Round 2
Reviewer 1 Report
minor commnets
1. (L56.)"internal environment" What does it mean?
2. (L192)"3.4 RNA-seq" is "3.3". Also, check L209.
Author Response
Response to Reviewer 1 comments
Point 1
- (L56.)"internal environment" What does it mean?
Response: Thank you very much for your comment. The internal environment (extracellular fluid) is the environment in which the cells live directly and the place where the cells exchange substances in a multicellular organism.
Point 2
- (L192)"3.4 RNA-seq" is "3.3". Also, check L209.
Response: Thank you very much for your comment.
L192: "3.4 RNA-seq" was changed to "3.3 RNA-seq".
L209: "3.3. KEGG enrichment analysis" was changed to "3.4. KEGG enrichment analysis".
